# Changes in the Histology of Lung Cancer in Northern Italy: Impact on Incidence and Mortality

**DOI:** 10.3390/cancers15123187

**Published:** 2023-06-14

**Authors:** Lucia Mangone, Francesco Marinelli, Isabella Bisceglia, Alessandro Zambelli, Francesca Zanelli, Maria Pagano, Giulia Alberti, Fortunato Morabito, Carmine Pinto

**Affiliations:** 1Epidemiology Unit, Azienda Unità Sanitaria Locale, IRCCS di Reggio Emilia, 42122 Reggio Emilia, RE, Italy; francesco.marinelli@ausl.re.it (F.M.); isabella.bisceglia@ausl.re.it (I.B.); 270064@studenti.unimore.it (A.Z.); 2Medical Oncology Unit, Azienda Unità Sanitaria Locale, IRCCS di Reggio Emilia, 42122 Reggio Emilia, RE, Italy; francesca.zanelli@ausl.re.it (F.Z.); maria.pagano@ausl.re.it (M.P.); giulia.alberti@ausl.re.it (G.A.); carmine.pinto@ausl.re.it (C.P.); 3Biotechnology Research Unit, Azienda Sanitaria Provinciale di Cosenza, 87051 Aprigliano, CS, Italy; f.morabito53@gmail.com

**Keywords:** lung cancer, incidence, mortality, morphology, survival, subtype

## Abstract

**Simple Summary:**

Lung cancer is a heterogeneous group of diseases in both morphology and molecular subtypes. There are two main histological categories of lung cancer: small-cell lung cancer (SCLC) and non-small-cell lung cancer (NSCLC). Although overall lung cancer mortality is declining in Italy, little is known about mortality trends by cancer subtype at the population level because death certificates do not record subtype information. This observational epidemiological study, using data from the Reggio Emilia Cancer Registry, evaluated the incidence, mortality, and survival trends of lung cancer according to the specific subtype. This study confirms the decline in the incidence and mortality of lung cancer in males, mainly squamous cell forms, inversely associated with cigarette smoking and the introduction of new drugs since 2013.

**Abstract:**

This study assessed the incidence, mortality, and survival of lung cancer subtypes of NSCSLC (non-small-cell lung cancer), SCLC (small-cell lung cancer), and other morphologies. It is an observational epidemiological study using 7197 cases from the Reggio Emilia Cancer Registry recorded between 2001 and 2020 in males and females. The incidence of NSCLC in 5104 males indicates a significant 3% annual increase until 2013 and then a decline of −3.2% that is not statistically significant; until 2014, mortality increased significantly (3.2%), but it then decreased non-significantly (−12.1%), especially squamous cell cancer. In 2093 females, the incidence and mortality trends continued to rise significantly through 2012, and then they began to slightly decline (not statistically significant). The two-year relative survival of NSCLC increased from 32% to 38% in males and from 42% to 56% in females. SCLC in males decreased significantly both in incidence and mortality, while in women, it showed a slight increase (significantly for incidence but not for mortality). This study is important because it analyzes the decrease in lung cancer in males and the increase in females in relation to the different histotypes. Our study’s findings confirmed a decline in male incidence and death beginning in 2013. We were unable to determine if the drop in cigarette smoking and the introduction of new drugs such as EGFR in first-line therapy were responsible for the lower incidence.

## 1. Introduction

Lung cancer is a heterogeneous group of diseases in both morphology and molecular subtypes [1]. There are two main histological categories of lung cancer: small-cell lung cancer (SCLC) (13% of cases) and non-small-cell lung cancer (NSCLC) (76% of cases) [2]. NSCLCs include a variety of histological forms, primarily adenocarcinoma and squamous cell carcinoma [3]. The diagnostic approach is deeply related to the development of personalized therapy and molecular and precise histological characterizations of lung cancer [4,5]. Lung cancer heterogeneity has implications in pathogenesis understanding, diagnosis, selection of tissue for molecular diagnosis, and therapeutic decisions. Understanding tumor heterogeneity is crucial for future developments [6]. Cigarette smoking is undoubtedly the most relevant risk factor for the onset of lung cancer; in fact, 85–90% of all lung cancers are attributed to it [7]. The risk increases with the number of cigarettes smoked and the duration of smoking [8]. The relative risk of smokers versus non-smokers increases approximately 14-fold and up to 20-fold in heavy smokers (over 20 cigarettes per day) [9]. Significant studies have also shown that stopping smoking substantially reduces the risk over time [10,11]. Unsurprisingly, the earlier smoking cessation is, the lower the chance of lung cancer, approaching the non-smokers’ incidence. Still, smoking cessation, even after many years, always follows a benefit in risk reduction [12]. Environmental or occupational exposures to radon, asbestos, and heavy metals, such as chromium, cadmium, and arsenic, also lead to an increased risk [13]. Exposure to airborne particulate matter and air pollution is classified by the IARC (International Agency for the Research on Cancer) [14] as a human carcinogen [15]. The ESCAPE study identified a significant increase in the risk of lung cancer of 22% for every 10 µg/m³ increase in PM 10 and 18% for every 10 µg/m³ increase in PM 2.5 [16]. Greater gains in risk are documented for adenocarcinoma, a lung cancer histological type more common in non-smokers [16,17]. In Italy, lung tumors with cyto-histological confirmation are approximately ¾ of the entire series (41,500 new cases); of these, 40% are adenocarcinomas (34% in men and 50% in women), 21% are squamous cell carcinomas (21% among men and 12% among women), 12% are from small cell tumors, 2% are from large cell tumors, and the remainder are from other and unspecified morphologies [18]. To date, a limited percentage of cases of NSCLC are diagnosed in the early stage [19] in which patients are potential candidates for surgery, possibly followed by chemotherapy to reduce the risk of recurrence or a locally advanced stage. In this setting, chemotherapy and radiotherapy represent the treatment backbone, possibly associated with immunotherapy at their completion [20,21]. In this respect, two critical therapeutic approaches have revolutionized the medical oncology therapeutic algorithm, i.e., molecularly targeted drugs and immunotherapy. Some drugs with a molecular target [22,23,24], a shining example among others, are the epidermal growth factor receptor (EGFR) inhibitors. In patients with the specific molecular target altered, these new drugs show a consistent clinical advantage to chemotherapy as the first-choice treatment. In the immunotherapy field, checkpoint inhibitors were first established as an effective treatment in relapse-resistant patients who had previously failed chemotherapy and subsequently proved to be superior to chemotherapy in first-line treatment, especially in patients with high PD-L1 expression in the neoplastic tissue [25,26]. Finally, in recent years, significant results have been obtained by combining chemotherapy and immunotherapy, even in cases with low or absent PD-L1 expression [27]. Although the incidence trends of the subtypes are well described [28,29,30], more knowledge is needed on the related mortality trends. More specifically, increasing awareness of lung cancer mortality trends according to histological subtype is extremely important to evaluate, since the potential adoption of lung cancer screening and smoking reduction can probably influence mortality differently related to the subtype [31,32,33].

This work aims to describe lung cancer incidence and mortality trends in a province of northern Italy by the histological subtype.

## 2. Materials and Methods

### 2.1. Study Setting

This observational epidemiological study used data from the Reggio Emilia Cancer Registry (RE-CR) approved by the provincial Ethics Committee of Reggio Emilia (Protocol no. 2014/0019740 of 04/08/2014). The RE-CR covers a population of 532,000 inhabitants and is considered a high-quality CR due to the up-to-date data that extend to the end of 2020 with a high percentage of microscopic confirmation (83.5% for lung cancer) and a rate of Death Certificate Only (DCO) below 0.1% [34].

### 2.2. Data Sources

The primary information sources of the RE-CR are anatomic pathology reports, hospital discharge records, and mortality data integrated with laboratory tests, diagnostic reports, and information from general practitioners. The study included lung cancer data divided into three groups as suggested by the International Classification of Diseases for Oncology, Third Edition (ICD-O-3) [35] as topography C34: Non-small-cell lung cancer (NSCLC), small-cell lung cancer (SCLC), and other morphologies. The morphological ICD-O-3 codes included in the NSCLC group were the following: 8140, 8250-8255, 8260, 8333, 8480-8481, 8490, 8550 (adenocarcinoma), 8070-8075, 8123 (squamous cell carcinoma), 8012-8014 (large cell carcinoma), 8021-8022 (anaplastic and pleomorphic carcinoma), 8031-8033 (giant cell and spindle cell carcinoma), 8046 (non-small cell carcinoma), 8200 (adenoid cystic carcinoma), 8240-8241/8243-8246/8249 (carcinoids), 8430 (mucoepidermoid carcinoma), and 8560 (adenosquamous carcinoma). The SCLC group included codes 8041-8045 (small cell carcinoma) and 8002 (malignant small cell tumor). Finally, the codes of the other morphologies group were 8000-8001 (malignant neoplasm), 8010 (carcinoma), 8020 (undifferentiated carcinoma), 8230 (solid carcinoma), 8800 (sarcoma), 8815 (fibrous tumor), 8890 (leiomyosarcoma), 8980 (carcinosarcoma), 9040 (synovial sarcoma), 9120 (hemangiosarcoma), and 9133 (epithelioid hemangioendothelioma).

### 2.3. Statistical Analysis

Descriptive analyses of patient characteristics, sex, age of diagnosis, year of diagnosis, and method of diagnosis were performed by groups. The standardized incidence and mortality rates of the last twenty years (2001–2020) were calculated for all three groups divided by males and females. For NSCLC, adenocarcinoma and squamous cell carcinoma were separately described. Population estimates, which were used to derive the rates, are represented by the general population of the Province of Reggio Emilia recorded on January 1st of each year. The incidence and incidence-based mortality rates were adjusted to the 2013 European standard population and calculated per 100,000 person-years. Finally, we calculated 2-year relative survival among patients with lung cancer according to sex, groups (NSCLC and SCLC), and calendar year using the relative survival approach and Pohar Perme method [36]; 95% confidence intervals (CI) were also reported, and a *p*-value ≤ 0.05 was defined as statistically significant. The trends over time were analyzed by calculating the annual percent change (APC) in age-standardized rates using joinpoint regression [37]. The APC is one way to characterize the trends in cancer rates over time; the output includes the estimated annual percentage rate change. With this approach, the cancer rates are assumed to change at a constant percentage of the rate of the previous year. As for the intervals used, joinpoint fits the selected trend data (incidence and mortality rates) into the simplest joinpoint model that the data allow. It is not always reasonable to expect that a single APC can accurately characterize the trend over an entire series of data. The joinpoint model uses statistical criteria to determine when and how often the APC changes. The maximum number of joinpoints predicted for these analyses was fixed to four. Analyses were performed using STATA 16.1 software (Stata Corp, College Station, TX, USA).

## 3. Results

The comparison of lung cancer deaths recorded on the death certificates (the death certificate is filled in by the necroscopist on a special form that certifies the fact of death and the cause of death as well as all the personal data of the deceased person) and cancer registry-based mortality does not show any significant variation in the 20 years considered (Figure 1), suggesting a good coding of the causes of death in the mortality registry. From 2001 to 2020, 7197 lung cancers were registered (5104 in males and 2093 in females) (Table 1). Most cancers (3082) were recorded at age 75+ years, 2356 at age 65–74 years, and 1264 at age 55–64 years. Less than 500 cases were registered under the age of 55 years. In the four periods considered (2001–2005, 2006–2010, 2011–2015, and 2016–2020), a different distribution of tumors by year of diagnosis is not observed, while the method of diagnosis shows approximately 80% of tumors with microscopic confirmation. Regarding the morphology, 4038 (56.1%) are NSCLC, 876 (12.2%) are SCLC, and 2283 (31.7%) are other morphologies.

Figure 2 shows the results for NSCLC separately for males and females. Among males (Figure 2, left), the incidence significantly increased by 3% annually until 2013 (95%CI 1.2; 4.9) and then non-significantly decreased (−3.2; 95%CI −7.5; 1.3). Mortality also significantly increased until 2014 (3.2; 95%CI 2; 4.4) and continued to non-significantly decrease in subsequent years (−12.1; 95%CI −32.0; 12.7). In females (Figure 2, right), the incidence significantly increased up to 2012 by 8.7% annually (95%CI 5.3; 12.3), while in subsequent years, the incidence remained stable: −0.7 (95%CI −5.3; 4.1). Mortality followed the same trend with a significant increase of 10.6% (95%CI 8.1; 13.1) until 2014 and then remained stable with a weak non-significant increase of 1.6% (95%CI −4.7; 8.2).

Adenocarcinoma showed a significant increase in the incidence in males up to 2012 of 7.0% (95%CI 3.6; 10.5), while in the subsequent period, the incidence was stable. Mortality continued to non-significantly increase by 3.7% (95%CI 2.3; −5.1) (Figure 3A). In females, the incidence showed a significant increase of 11.7% (95%CI 7.9; 15.6) until 2011 followed by a substantial significant increase in mortality of 11% (95%CI 8.2; 13.9) until 2015. In subsequent periods, the incidence and mortality showed a slight downward non-significant trend (Figure 3B). In regards to the squamous cell trends, on the other hand, there was a significant decrease of incidence by −2.4% (95%CI −4.2; −0.5) followed by a significant decrease in mortality of 4.0% in males (95% CI −4.9; −3.0) (Figure 3C), while in females, both trends were slightly non-significantly increased (Figure 3D).

In regards to SCLC in males, there was a significant decrease in the incidence of −2.1% (95%CI −4.0; −0.2) and mortality of −1.7% (95% CI −3.4; −0.1); in females, on the contrary, the incidence significantly increased by 2% annually (95%CI 0.4; 3.7), while mortality showed a slight non-significant increase of 0.5% (95%CI −2.5; 3.5) (Figure 4).

The two-year relative survival among patients with NSCLC was higher among women than among men; survival among women improved from 42% (95%CI 25; 59) in 2001 to 56% (95%CI 44; 67) in 2018 (Figure 5A), while in males, it passed from 32% (95%CI 21; 42) to 38% (95%CI 30; 46) in the same period (Figure 5B).The survival among men patients with SCLC, remained constant over the years (from 11% (95%CI 4; 25) in 2001 to 13% (95%CI 5; 31) in 2018) (Figure 5C), while in women, a slight increase (from 15% (95%CI 3; 37) to 18% (95%CI 5; 39) in the same period was observed (Figure 5D).

In regards to the category of other morphologies (Figure 6), in males, after an initial non-significant increase in incidence up to 2004, a significant decrease of −14.7% (95%CI −19.4, −9.7) was observed up to 2012 followed by a modest reduction in subsequent years. Mortality showed a similar trend with a slight non-significant increase up to 2007 and a significant decrease of −29% (95% CI −48.4; −2.3) followed by a slight non-significant decrease of −2%. In females, there was a significant increase in incidence up to 2007 of 6% (95%CI 0.1; 12.2) followed by a significant decrease of −30.5% (95%CI −51.5; −0.5) and, then again, a significant increase of 5.4% (95%CI 1.4; 9.6) followed by mortality, which had completely overlapping trends.

## 4. Discussion

This study described the trends in mortality among patients with a different subtype of lung cancer in a northern Italian province and showed an increase in incidence and mortality in males first in 2013 and then a subsequent decrease after that year for NSCLC. In females, on the other hand, there was a constant increase in both indicators. As for SCLC, there was a constant decrease in the incidence and mortality in males, while in females, the opposite occurred, especially for the incidence. In general, survival is increasing for both histological types and genders, although it is more marked in females in the NSCLC form.

In general, the incidence rates in the USA fall more in metropolitan than non-metropolitan areas, more in males than females, and more in the young than in the elderly [38]. The variability in the occurrence of lung cancer may be explained by many risk factors that were not investigated in this study [39,40,41].

In the UK, the relative risk of lung cancer is 14 times higher in smokers than in non-smokers. In the USA, the mortality from lung cancer is 101.2 per 100,000 smokers and 8.6 per 100,000 non-smokers. In addition, the risk of cancer is 2.6 and 2 times greater in smokers and female smokers in China, respectively [42]. In America and Europe, the APC (annual percent change) is decreasing in males (from −0.7 in 1975–1985 to −2.9 in 1985–2005) and increasing in females (from 1.3 to 5 in the same period); in Asia, the trends are stable. The morphology of adenocarcinoma is more frequent in the black population in the USA and China (the lowest in India), while adenosquamous is frequent in Slovakia, Germany, and France [43]. The trends appear to be decreasing in males and are also beginning to drop in females in many countries. The incidence increased from 100 to 130 cases per 100,000 males in 1980 before declining to 80 cases per 100,000 in 2005 in Singapore. In females, the rate also reached 40 cases per 100,000 before falling to 25 per 100,000 [44]. In the USA, between 2005 and 2009, the incidence was higher in males than in females, except for young people under the age of 45 years. In both sexes, the trend is decreasing both in men (APC −2.6) and women (APC −1.1) [45]. Recent work showed that NSCLC mortality decreased faster than the incidence, and this was associated with a substantial improvement in survival coinciding with the approval of target therapies. In males, the incidence-based mortality of NSCLC decreased by 6.3%, while the incidence decreased by 3.1% annually, related mainly to the EGFR first-line therapy introduction in 2013 [2]. Similar patterns were observed in females with NSCLC. In contrast, mortality from NSCLC declined, but the decline is related almost entirely to a decrease in the incidence rather than an increase in survival.

Unlike what was stated in Hawlader’s paper [2], we did not observe substantial differences in terms of mortality incidence comparing deaths recorded by the Death-Certified Mortality database and Cancer Registry over the 20 years examined. This confirms the optimal quality of our Cancer Registry data and the coding of mortality in general. For NSCLC in males, the increase in incidence until 2013 was followed by a decline in incidence (although not significant) until recent years. Mortality followed a similar pattern, first rising and then falling with period overlap. The trend is very similar to what was reported by the American study, even if the small numbers in our case do not allow for reaching statistical significance. Similar patterns are observed in women: a sharp increase in incidence and mortality followed by a plateau phase. The two-year survival shows an almost constant trend over the years, approximately 32% in males with an increase to 38% for cases diagnosed in 2018; in females, survival increased from 42% in the first few years to 56% in 2018 with improvements similar to what is reported in other studies [2,46]. Specifically, in our study, an increase in adenocarcinoma, both in terms of incidence and mortality, was recorded in males; the significant decrease was only the prerogative of the squamous cell forms. In addition, our current paper shows that the drop in mortality (−4.0) is almost double compared to the reduction in the incidence (−2.4%) observed in Hawlader’s study. We can only speculate that the decrease in the incidence and mortality of squamous forms may be somewhat attributed to the decline in cigarette smoking among men because there are no data on this issue available for our research [47]. Still, it is also due to the introduction of new drugs starting from 2011 with tyrosine kinase inhibitors in EGFR-mutated adenocarcinoma and from 2017 with immune checkpoint inhibitors [48].

Also, the SCLC forms show a significant decrease in incidence and mortality in males, as in the study by Hawlader [2]. In contrast, the incidence increases in females, but the mortality remains constant. On the other hand, survival has not been affected in the slightest by medical progress, settling at values of approximately 11–12% over the years in males. Unfortunately, too small numbers were accounted for in females, preventing any analysis. The other forms group is usually excluded from the analyses, although representing roughly one-third of cases, and deserves few comments. In this setting, a significant and continuous decrease in the incidence in males was detected. More importantly, we documented a decrease in mortality (−29%), suggesting that these are NSCLC forms for which treatment innovation has positively and powerfully impacted the survival length. In females, the significant increase in incidence over the years coincided with a decrease in mortality over the same period, although not significant.

Among the strengths of this study, we should mention the availability of 20 years of registration; the good quality of the data not only on incidence but also on mortality; the availability of very recent data; and the possibility of stratifying NSCLC into adenocarcinoma, squamous cell, and other morphologies. A limitation of this study is related to the fact that the numbers are relatively small since they refer to a single Cancer Registry. Another limitation of this study is the absence of individual information on the smoking habits of the cases in our study; therefore, only correlations with smoking habit trends in Italy can be made [49]. In addition, we do not have individual data on the type of treatment; therefore, we can only associate the introduction, for example, of the drugs used since 2014 in our province.

## 5. Conclusions

Based on a sample of more than 7000 patients with lung cancer in a province of northern Italy, our study confirmed a decline in male incidence and mortality since 2013 for NSCLC and for the squamous cell form in particular. However, we were unable to determine whether the decline in cigarette smoking and the introduction of new drugs such as EGFR into first-line therapy were responsible for the lower incidence [50]. In females, on the other hand, we noticed an increase in adenocarcinoma. Two-year survival for both sexes is increasing. For SCLC, mortality and incidence are decreasing in males, while they are slightly increasing in females.

## Figures and Tables

**Figure 1 cancers-15-03187-f001:**
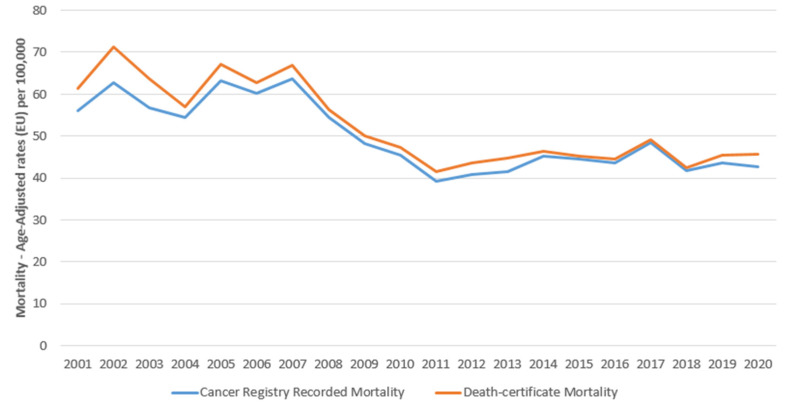
Reggio Emilia Cancer Registry. Years 2001–2020. Mortality Estimates Based on Data from Death Certificates and on Cancer Registry Recorded Mortality with Lung Cancer.

**Figure 2 cancers-15-03187-f002:**
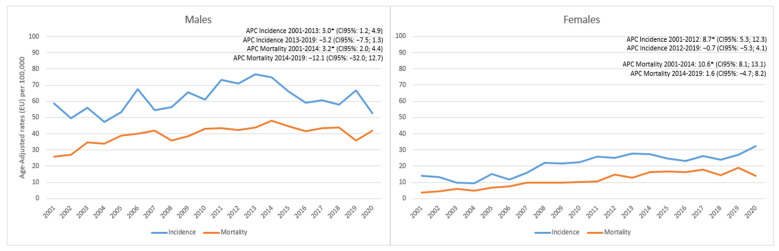
Reggio Emilia Cancer Registry. Years 2001–2020. Non-small cell lung cancer (NSCLC) incidence and mortality trends among males and females. * APC is significantly different from zero at the alpha = 0.05 level.

**Figure 3 cancers-15-03187-f003:**
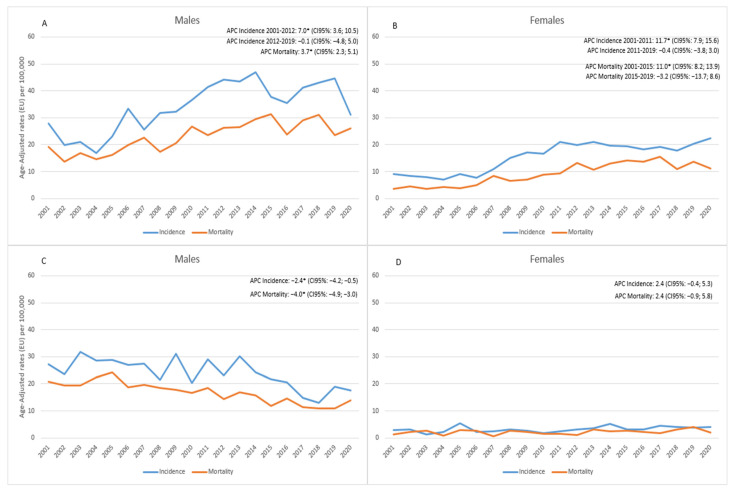
Reggio Emilia Cancer Registry. Years 2001–2020. Adenocarcinoma (**A**,**B**) and squamous cell carcinoma (**C**,**D**) incidence and mortality trends among males and females. * APC is significantly different from zero at the alpha = 0.05 level.

**Figure 4 cancers-15-03187-f004:**
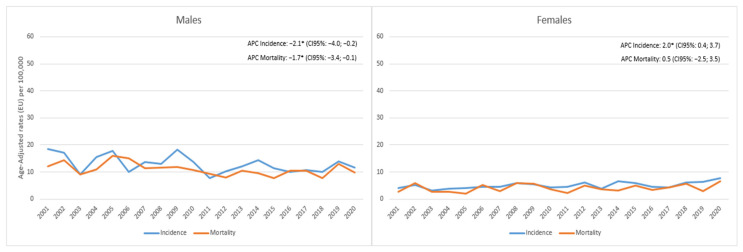
Reggio Emilia Cancer Registry. Years 2001–2020. Small-cell lung cancer (SCLC) incidence and mortality trends among males and females. * APC is significantly different from zero at the alpha = 0.05 level.

**Figure 5 cancers-15-03187-f005:**
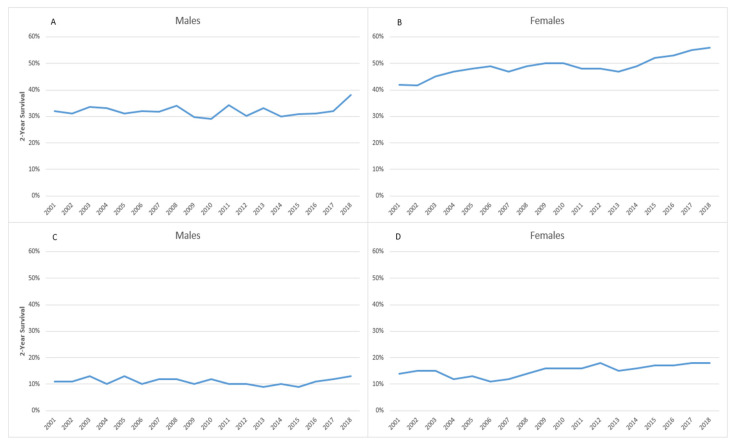
Reggio Emilia Cancer Registry. Years 2001–2018. 2-Year survival trends in NSCLC (**A**,**B**) and SCLC (**C**,**D**) among males and females.

**Figure 6 cancers-15-03187-f006:**
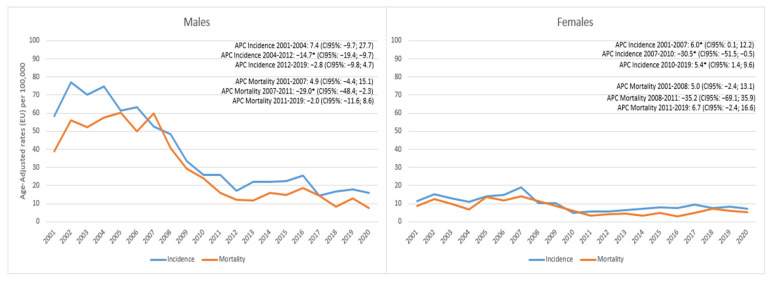
Reggio Emilia Cancer Registry. Years 2001–2020. Other morphologies of lung cancer incidence and mortality trends among males and females. * APC is significantly different from zero at the alpha = 0.05 level.

**Table 1 cancers-15-03187-t001:** Reggio Emilia Cancer Registry. Years 2001–2020. Number of new cases by sex, age, year of diagnosis, and method of diagnosis.

	Small Cell	Non-Small Cell	Others	Total
	*n* (%)	*n* (%)	*n* (%)	*n* (%)
Overall	876 (12.2)	4038 (56.1)	2283 (31.7)	7197 (100.0)
Sex				
Males	596 (68.0)	2867 (71.0)	1641 (71.9)	5104 (70.9)
Females	280 (32.0)	1171 (29.0)	642 (28.1)	2093 (29.1)
Age at diagnosis				
15–54	62 (7.1)	332 (8.2)	101 (4.4)	495 (6.9)
55–64	175 (20.0)	847 (21.0)	242 (10.6)	1264 (17.6)
65–74	330 (37.6)	1489 (36.9)	537 (23.5)	2356 (32.7)
75+	309 (35.3)	1370 (33.9)	1403 (61.5)	3082 (42.8)
Year of diagnosis				
2001–2005	213 (24.3)	690 (17.1)	866 (37.9)	1769 (24.6)
2006–2010	219 (25.0)	937 (23.2)	652 (28.6)	1808 (25.1)
2011–2015	211 (24.1)	1244 (30.8)	379 (16.6)	1834 (25.5)
2016–2020	233 (26.6)	1167 (28.9)	386 (16.9)	1786 (24.8)
Method of diagnosis				
Histological	684 (78.1)	3373 (83.5)	375 (16.4)	4432 (61.6)
Cytological	192 (21.9)	660 (16.4)	425 (18.6)	1277 (17.7)
Clinical/instrumental	0 (0.0)	5 (0.1)	1483 (65.0)	1488 (20.7)

## Data Availability

The data presented in this study are available on request from the corresponding author. The data are not publicly available due to ethical and privacy issues; requests for data must be approved by the Ethics Committee after the presentation of a study protocol.

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
