# Peer review of "Changes in the Histology of Lung Cancer in Northern Italy: Impact on Incidence and Mortality"

_cancers, 2023, doi:10.3390/cancers15123187_

Round 1

Reviewer 1 Report

This is a descriptive study that documents the trends in mortality due to lung cancers in northern Italy.  It adds to the literature on lung cancer mortality by focusing on histological subtypes of lung cancer.  The authors utilize data gathered from a well-regarded cancer registry, the Reggio Emilia Cancer Registry, which covers over half a million inhabitants and has microscopic confirmation of cancer type for over 80% of lung cancers and highly complete death certificate data.

Overall, I think their trend analyses are done appropriately.  I made a few recommendations.  I described a few places where things could be clearer from a technical point of view.  My main concern is with their Discussion of their findings which I had a hard time connecting to their results. 

The authors conclude that “The results confirm a decrease in the incidence and mortality in males starting from 2013…related both to the decline in the incidence due to the decrease in cigarette smoking and to the acquisition of new drugs, i.e., EGFR, in first-line therapy”.  I do not find this warranted by their results.  They are making a leap between their observed trend and the causes of changes in directions.  While the identified possible causes may be backed by the literature, they did not test hypotheses regarding the effect of things like new drugs in this study.  They need to be much clearer, and show a bit more humility, about the connections they are making, how they are making them, and what the results of their descriptive study actually show.   

Methods

I believe the STATA software package needs a reference.  See the Stata website.   In the Results the authors pick varying cut points in time when describing the trends and changes in them.  The rationale for this should be presented in the Methods section.

Results

Figure 1.  The labels could be clearer.  I believe both lines indicate mortality rates recorded in two different sources.  Instead of “Cancer Registry” say something like “Cancer Registry Recorded Mortality.”  Also, add “Mortality” to the Y-axis label.  I would also indicate in the text that “death certificates” means vital statistics data, or whatever the name of the source is in this region of Italy.  It would just make it a bit easier to be sure about what the authors are telling us. 

I might interpret “a slight decline by 138 -0.7 (95%CI -5.3; 4.1)” as basically flat, especially since the author follow this by interpreting “a weak non-significant increase of 1.6% (95%CI -4.7; 8.2).” 

Is it possible to get confidence limit estimates on the improvements in 2-year survivorship? 

Why do the authors shift cut points in the trends they report in the Results?  They seem to imply early in the manuscript (in the summary and the abstract) that some change in medication in 2013 led to reductions in mortality.  Why move this date to 2012 and 2015 when describing various trends?  I understand that this is not analytic epidemiology, and the authors are not testing hypotheses, but then this means they need to be very careful about associating declines in mortality to the advent or widespread use of particular cancer treatments. 

Line 156 is in Italian and needs to be translated. 

Discussion

The first paragraph of the Discussion wanders all over the place, describing a wide range of risk factors for cancer including “earthquake marginalization”???  It’s difficult to see any connection between the trends they report in their results and these factors listed in the Discussion.  This needs to be much more focused.   

The author report “In America and Europe, the APC (Annual Percent Change) is decreasing in 206 males (from -0.7 to -2.9) and increasing in females (from 1.3 to 5); in Asia, the trends are 207 stable.”  During what time period are they referring?  And how do they connect this to their findings?  This is followed by several more various cancer demographics among several countries.  More focus, please. 

“Unlike what was seen in Hawlader's work, where the incidence-based mortality showed a more accentuated figure since Death-certified mortality has a bias linked to deaths from lung metastases” is an incomplete sentence.  And after rewriting it (or at least connecting it to the sentence prior to it), add a reference to Hawlader’s work. 

The authors finally get to discussing their findings around line 220.  I suggest the authors heavily edit the Discussion up to this point perhaps eliminating much of it.  I suggest expanding the discussion of the relevance of their findings. 

What is a “double drop in mortality (-29%)”?  A double digit drop? 

Finally, when the authors conclude that “The results confirm a decrease in the incidence and mortality in males starting from 2013, as observed in the SEER study, related both to the decline in the incidence due to the decrease in cigarette smoking and to the acquisition of new drugs, i.e., EGFR, in first-line therapy” they are making a leap between their observed trend and the causes of changes in directions.  While the identified possible causes may be backed by the literature, they did not test hypotheses regarding the effect of things like new drugs.  They need to be much clearer about the connections they are making, how they are making them, and what their results actually show in this descriptive study. 

No major concerns.  A few comments and suggestions/corrections were noted in the general comments to authors.  

Author Response

Dear reviewer,

thank you for your comments! We hope that our answers conform to tour requests.

Best regards,

Lucia Mangone

Reviewer 2 Report

The manuscript (ID: cancers-2397807) aimed to describe the incidence, mortality, and survival of main histological categories of lung cancer (non-small-cell lung cancer, small-cell lung cancer, and other morphologies) from the northern Italy between 2001 and 2020, in both genders. Work corrections are needed (major revision):

  • Lines 19-29: Reconstruct the Abstract, in such a way as to clearly state the background of this study, the objectives of the study, the study design used, with a brief description of the applied statistical methodology. Pay special attention to Lines 26-29, because such a claim must be made only on the basis of one's own results.
  • Lines 19-29: In the presentation of the results in the Abstract, it is mandatory to indicate which changes in trends were statistically significant and which were not statistically significant.
  • Mauscript as a whole: Correct words such as `We assessed …`, `... we must know …`, `… we also split …`, `… we provide estimates …`, `… we reported …`, ` ... we describe ...`, `... we observed ...`, `... we documented ...`, `... we must mention ...`, using a writing method that corresponds to a publication in a scientific journal.
  • Lines 54-56: In this sentence, enter references for the ESCAPE study, that is, reference No. 17, according to the reference list at the end of this paper. Align the order of references in the list of references with the order of references in the text of the paper.  
  • Lines 85-121: It is necessary to make the Methods section more clear and transparent. Enter subsections:  
    • Study design. Check whether the correct study design applied in this work is entered on Line 86. Correct in the Methods section (Line 86), and Line 280;
    • Study setting;   
    • Study population;    
    • Data sources:      
      • Present data on the quality of the registry.
    • Measures;    
    • Statistical analysis: define APC, provide appropriate references.
  • Lines 141-144: The mentioned data about survival are not shown on `(Fig. 2b)`. Check and correct.
  • Lines 133-165: In that text in the Results section, the corresponding figures are missing in the appropriate place in the text. To correct.
  • Lines 156-157: Check and correct.    
  • Line 187: Insert a new paragraph in the Discussion section, as the first paragraph, in which the most important results of this study will be highlighted.     
  • Lines 188-258: The Discussion chapter must be completely reconstructed. Above all, give a clear and logical flow to the text of this section, and in accordance with the presented own results. Correct as follows:  
    • The focus must be on comparing own results with data from similar studies in other regions or countries;     
    • Give a possible explanation for the differences in incidence, mortality and survival for lung cancer according to the characteristics shown in this manuscript (age-standardized rates of incidence and mortality, trends, males/females, morphologies, survival);  
    • Give a possible explanation for the described trends in incidence and mortality in this region;    
    • Many sentences are missing appropriate references. To correct.      
  • Lines 259-265: In the paragraph on limitations of the study, discuss, as limitations of this study, the study design that was applied in this research, the lack of data on therapeutic modalities and data on exposure to risk factors at the individual level, etc.    
  • Lines 266-271: Reconstruct the Conclusions, in a way to summarize and highlight only the most important results of this study, without unnecessary comparisons with the SEER study, nor any references. The place for such data is in the Discussion section. To correct.  

The quality of English language is appropriate.  

Author Response

Dear reviewer,

thank you for your comments, we hope that our answers conform your requests.

Best regards,

Lucia Mangone

Reviewer 3 Report

Authors assessed the incidence, mortality, and survival of lung cancer subtypes of NSCSLC (non-small-cell lung cancer), SCLC (small-cell lung cancer), and Other morphologies using 7,197 cases from the Reggio Emilia Cancer Registry recorded between 2001 and 2020 in males and females.  They reported the decrease in the incidence and increase in the survival.  They however write that their study shows the connection between a decline
in cigarette smoking, the introduction of new drugs starting in 2013, and the decline in the incidence and mortality of lung cancer in males, primarily of the squamous cell cancer.    The problem is that authors do not present the data on
decline in cigarette smoking.  How can readers be sure that decrease in the cancer incidence is really associated with decline in cigarette smoking? 

Author Response

Dear Reviewer 3,

thank you for your comment, we hope that our answers conform to your requests.

Best regards,

Lucia Mangone

Round 2

Reviewer 2 Report

Thank you for the opportunity to re-review the manuscript (ID: cancers-2397807). The authors made certain corrections, answered most of my comments, but did not answer all of the questions or provide satisfactory explanations.

As a whole, the revised version of the work, without the introduction of important corrections, remains a large number of scattered data, on the basis of which the authors did not highlight the most important results.

Major comments:

  • First, the most important issue is `Study design`, in the Methods section: Study design is not corrected. A cohort study design was not applied in this paper, as is written in both versions of this manuscript. In this work, an observational epidemiological study was conducted, by the type of descriptive study. Correct this.
  • In the Statistical analysis subsection, no necessary changes were made in connection with the joinpoint regression analysis, on which almost all the results in this paper are based. Namely, the Measures were not described, and it was necessary to: specify the definition of APC, the maximum number of joinpoints that the authors predicted with the models of this analysis, define `significantly increased` trend, `significantly decreased trend`, `non-significantly increased` trend, `non- significantly decreased trend`, and `stable` trend, and apply throughout the work. 

Minor comments:

  • My comment on the Abstract did not mean the introduction of a structured version of the abstract - it is an arbitrary interpretation of the authors of this work. It was necessary to briefly say why the topic covered by the paper is important, and especially to accurately and precisely state the study design that was applied in this paper.
  • In the Abstract (Lines 30-32) it is unnecessary to insert the text from the Conclusion section from the previous version of the paper: namely, neither a comparison with the SEER database was carried out in the processing of the data in this paper, nor was a correlation study carried out in this paper in order to state in the conclusion what kind of link with any factors. Correct in such a way as to focus on one's own results.
  • The paragraph on limitations should be improved further.
  • The Conclusion section again provides a comparison with the SEER database, which was not analyzed in this paper. The same applies to specifying determinants, e.g. smoking, etc. Summarize and highlight your own most important results.

The quality of English language is appropriate. 

Author Response

Dear Reviewer,

thank you for your comments! We hope that our answers satisfy your requests.

Best regards,

Isabella Bisceglia

Reviewer 3 Report

N/A

Author Response

Dear Reviewer,

thank you for your comments, we hope that our answers satisfy your request.

Best regards,

Isabella Bisceglia

Round 3

Reviewer 2 Report

Thank you for the opportunity to re-review the manuscript (ID: cancers-2397807) again. The authors made numerous corrections, except that the study design was not specified correctly. The design of the study applied in this paper is an observational epidemiological study, that is, the descriptive study, which describes the data of the regional register. Correct this.  

The quality of English language is appropriate. 

Author Response

Dear reviewer,

thanks for your comment. We have corrected it in all the text.

Best regards,

Lucia Mangone